# Finite Element Simulation of Punch Structure and Its Effect on Microstructure Evolution of Mg-Gd-Y-Zn-Zr Alloy via Rotary Extrusion Method

**DOI:** 10.3390/ma15155248

**Published:** 2022-07-29

**Authors:** Lin Yan, Beibei Dong, Zhimin Zhang, Yong Xue, Mei Cheng

**Affiliations:** College of Materials Science and Engineering, North University of China, Taiyuan 030051, China; yl_nuc2019student@163.com (L.Y.); zhangzm_nuc_pro@163.com (Z.Z.); xueyong_nuc@163.com (Y.X.); chengmei_nuc@163.com (M.C.)

**Keywords:** finite element simulation (FEM), gradient structure, Mg-Gd-Y-Zn-Zr alloy, microstructure, rotary extrusion (RE)

## Abstract

This article aims to explore the impact of the punch structure (number of grooves, area ratio of grooves, depth of grooves and flaring angle) on the loading, torque and metal flow during the rotary extrusion (RE) process via finite element simulation (FEM) software. In order to further verify the simulation results, physical experiments were carried out and the microstructure of Mg-Gd-Y-Zn-Zr alloy after RE deformation was characterized and analyzed. The FEM results indicated that increasing the groove number will increase the amount of shear deformation and promote the metal at the bottom of the punch to flow rapidly to the cylinder wall. The increase in the groove depth would continue to reduce the forming load and increase the strain. However, if the groove depth exceeded 6 mm, an excessive flow-velocity difference would be formed, resulting in the formation of folding defects. The time of metal flow from the bottom of the punch to the cylinder wall would be shortened with the increase in flaring angle. Therefore, a groove number of 8, an area ratio of 64.49%, a groove depth of 6 mm and a flaring angle ranging from 7° to 9° were the optimal parameters of the punch structure to form the Mg-Gd-Y-Zn-Zr cylindrical parts via the RE technique. In addition, the cylinder parts could be formed with good quality according to the optimized FEM results. The cylinder wall from inner region to outer region exhibited gradient microstructure owing to the different metal flow and strain during the RE process.

## 1. Introduction

High-performance magnesium (Mg) alloys have received considerable research interests due to their desirable properties, such as low density, high strength, easy cycling, etc. in the transportation and aerospace fields [1,2,3,4]. The proper combinations of strength and ductility are the necessary conditions that should be satisfied before the industrial usages of Mg alloys [5,6,7]. Conventional plastic deformation methods have limited effects on grain refinement, second-phase distribution and texture of the alloy, resulting in limited mechanical properties of the alloy after deformation, and the degree of grain refinement is related to the size of deformation strain. Therefore, scholars have conducted in-depth research on introducing greater strain into the alloy deformation process.

The purpose of severe plastic deformation (SPD) technology is to carry out single or multiple deformation passes on the alloy to achieve high strain, so as to make the alloy have superior mechanical properties. SPD technology makes a significant contribution to grain refinement, and the grain size can reach the micron or even nanometer level. SPD methods not only have advantages in grain refinement, but also have a significant effect on the uniform distribution of the second phase of fine particles and the formation of deformation texture, which can greatly improve the strength and toughness of the alloy. It provides a strong support for the development of strengthening and toughening of Mg alloys. At present, SPD technologies that have been researched and developed include repetitive upsetting-extrusion (RUE) [8,9,10], multidirectional forging (MDF) [1,2,11], equal channel angular pressing (ECAP) [12,13,14], high pressure torsion (HTP) [15,16,17] and so on. Xu et al. [10] investigated AZ61 alloy prepared by repetitive upsetting and extrusion, and found that the average grain size of the alloy can be significantly refined to 3.5 µm after 3 passes at 558 K. Dong et al. [2] found that the average grain size of Mg-13Gd-4Y-2Zn-0.5Zr alloy decreased significantly and increased slightly with the increase in MDF passes under decreasing temperature from 480 °C to 420 °C. Jahadi et al. [14] conducted ECAP technology to extruded AM30 bar at 275 °C, the results showed that the grain refinement of the alloy after deformation was significant, and the average grain size was refined from 20.4 µm to 3.9 µm. Harai et al. [17] found that the average grain size of AZ61 Mg alloy could be achieved to 100 nm and exhibited superplasticity via the HPT method.

In addition, Mg rare-earth (RE) alloys have developed from the Mg-Th system and Mg-Y system to Mg-Gd system, and WE54 and WE43 system alloys are high-strength heat-resistant Mg alloys with mature commercialization so far [18]. Rokhlin et al. [19] found that the solid solubility of Gd in Mg matrix was greater than that of Y, and the strengthening of Mg alloys was more obvious than that of Y, Nd and other RE elements. Drits et al. [20] found that the mechanical properties of Mg-22wt.%Gd alloy after T6 treatment were up to 400 MPa, but its elongation was low, and its poor plasticity was an obvious defect. Adding Y to Mg-Gd system alloys can properly refine the coarse grains of as-casted alloys. The addition of Y has a certain effect on improving the mechanical properties of as cast alloys. The ultimate tensile strength (UTS) and tensile yield strength (TYS) of Mg-7Gd-5Y (wt.%) alloy at room temperature reached 258 MPa and 167 MPa [21]. The Mg-9Gd-4Y-0.5Zr alloy prepared by Hong et al. [22] exhibited excellent mechanical properties, and its UTS, TYS and elongation (EL) reached 508 MPa, 400 MPa and 8.0%, respectively. Xu Chao et al. [23] prepared Mg-8.2Gd-3.8Y-1.0Zn-0.4Zr alloy with UTS of 505 MPa, TYS of 416 MPa and EL of 12.8% through extrusion technology. Different from the results by Hong et al., Zn element was added to the alloy, which further improved the strength of Mg alloy owing to the existence of long-period stacking ordered (LPSO) phases and second particles.

In recent years, more researchers have proposed a new SPD method to overcome the disadvantages of the traditional SPD methods requiring multi-passes to obtain large accumulative strain [24]. The new SPD method is rotary extrusion (RE), combining torsion and conventional extrusion (CE) [25]. At present, there are three kinds of RE studies at home and abroad: (i) There is no substantial difference between the punch-and-die cavity and traditional extrusion die. In the process of punch-or-die rotation, the purpose of deformation is achieved by relying on surface friction, which is similar to HPT [26,27,28]; (ii) The spiral structure is designed in the cavity of the concave model. During the extrusion process, the metal flows along the spiral structure in order to extend the metal deformation path and produce the torsion effect. In this process, the strain is accumulated to increase the deformation [29,30]; (iii) The shearing structure is designed on the working surface of the punch or the inner bottom surface of the female die, and the RE deformation is completed by introducing the rotation factor into the die. This method focuses on the design of the shearing structure, which will directly affect the strain and uniformity of the overall extrusion deformation [24].

The shear structure parameters at the bottom of the punch are very important. The determination of the number (area ratio), depth of the groove and the flaring angle are the key factors to whether the RE deformation can be well-formed, and these parameters need to be determined by analyzing the gradient simulation results. Finite element simulation (FEM) can quickly optimize the reasonable die structure, shorten the research cycle and improve the success rate of the experiment [31]. Therefore, the paper aims for the finite element simulation (FEM) of punch structure, and its effect on microstructure evolution of Mg-Gd-Y-Zn-Zr alloy via RE method are studied.

## 2. Materials and Methods

### 2.1. Finite Element Simulation (FEM) of Punch Structure

The general idea of RE deformation research in this paper is RE-parameters simulation-experimental verification, as shown in Figure 1. The structure of the petal groove RE die was determined through simulation and optimization, and the Mg-Gd-Y-Zn-Zr alloy was used for preliminary exploration of RE. The FEM was carried out by Deform software, and Figure 2 shows the RE forming experimental equipment and die. In addition, FEM and optimization for petal groove punch were carried out to determine its structural parameters. Gradients were set for the number of grooves (groove area ratio), groove depth and flaring angle in revolution for simulation and analysis. The gradient values of each parameter are shown in Table 1 and the schematic diagrams of the groove depth and flaring angle and FEM punch models are shown in Figure 3 and Figure 4, respectively.

In the FEM process, the blank size was 260–100 mm (φ, h). The inner diameter of the female die was 280 mm, and the diameter of the working belt of the male die was 220 mm. The three-dimensional (3D) model of the die and blank was designed by NX software according to the size, and imported into DEFORM-3D software in stl. file format. The H13 die steel provided by the system was selected as the die, and the Mg alloy material model created by the laboratory was selected as the blank. The blank was meshed by tetrahedral elements; the number of meshes was 100,000, and the number of nodes was 29,351. At the same time, mesh rezoning and volume compensation were selected to reduce the calculation error caused by mesh distortion in the deformation process. The critical relative interference depth of mesh rezoning was set to 0.7. In the simulation process, it was set as isothermal deformation and obeys the von Mises yield criterion. The shear friction model was selected for the friction between the blank and the die, because generally, the shear friction was more suitable for metal volumetric forming, and the friction coefficient was set to 0.3. Sparse solver and Newton–Paphson iteration method are used to calculate the whole process of RE simulation. The material model, boundary conditions and friction contact settings are described in Section 2.1 of the article. In the simulation process, the step length is 1 mm/step, and the total stroke is 75 mm. The final simulation result with the stroke of 75 mm is selected as the analysis data.

The blank height was 100 mm, the total extrusion stroke was set as 75 mm, the longitudinal extrusion speed was 1 mm/s and the die rotation speed was 0.314 rad/s. As the petal-shaped die would rotate the blank in the process of rotation, to avoid the premature rotation leading to the tearing of the blank by torsion, it was first squeezed 30 mm longitudinally to ensure that the working belt of the punch fully entered the blank, and then the longitudinal feed of the punch and the circumferential rotation of the concave were carried out at the same time between 30 mm and 75 mm. In the simulation, the full stroke reverse extrusion of punch without groove was used as a comparative simulation.

### 2.2. Microstructure Characterization

The billet temperature was set as 420 °C and the die temperature was set as 450 °C, and the RE experiment was conducted on 1250 t-hydraulic press. The observation samples were cut from the initial alloy and the REed barrel for microstructure analysis, and the sampling positions are shown in Figure 2b. After the samples were polished from 400# to 7000# sandpapers, they were further mechanically polished by MP-2A polishing machine. Then, a mixed reagent of 1 g picric acid, 2 mL distilled water, 2 mL acetic acid and 25 mL ethanol was selected for etching. The microstructure observation was along the extrusion direction (ED) of the REed samples via optical microscope (OM, Zeiss Axio Imager-A2m, Germany) and scanning electron microscope (SEM, Hitachi-SU5000, Japan) to verify the FEM results. The brightness and exposure time were selected in the automatic mode, and the contrast was fine-adjusted according to the actual light. The accelerating voltage was 20 kV, the point strength was 50, and the working distance was 10 mm for microstructure observation by SEM.

## 3. Results

### 3.1. FEM Results and Analysis

The comparison of load and torque of FEM under the gradient of the number of grooves (area ratio) is shown in Figure 5. The number of grooves with 0 (0%) is reverse extrusion deformation, and the rest is RE deformation. In the first stage of initial deformation, the load increases continuously to the highest point in a short time due to the influence of work hardening. Then, the load decreases in a small range (the second stage of deformation), which is the softening of the alloy due to the dynamic recrystallization (DRX) behavior caused by deformation. As the deformation enters the third stage, the load region is relatively stable, and the work hardening and DRX softening tend to balance. After the stroke of 30 mm, the addition of rotary motion makes the deformation load drop suddenly, which also shows that RE deformation can effectively reduce the forming load. According to the simulation results in Figure 5a, the load after back extrusion is ~5750 kN, and the rotating extrusion load with the number of grooves (area ratio) of 2 (16.12%), 4 (33.30%), 6 (48.39%) and 8 (64.49%) are ~5000, ~4630, ~4300 and~4120 kN, respectively, which are reduced by 13.04%, 19.48%, 25.22% and 28.35% compared with the reverse extrusion deformation load. The punch has its own shear groove structure; when the punch drives the blank to rotate in a circumferential direction, the blank will produce shear deformation through the groove structure. Under this deformation mode, the combined action of friction and metal shear deformation resistance will produce a relative torque of 2.8 × 10^4^, 3.5 × 10^4^, 4.3 × 10^4^, 4.8 × 10^4^ N·m, corresponding to the groove numbers of 2, 4, 6 and 8, respectively.

First of all, it is certain that RE deformation can effectively reduce the forming load compared with traditional back extrusion deformation, which has been confirmed by relevant researchers [26,28]. Secondly, as the number of grooves (area ratio) increases in revolurion, the extrusion load continues to decrease, and finally the load decreases by 28.35% in the structure of eight grooves (64.49%), but the torque will increase.

The comparison of strain distribution under the gradient of the number of grooves (area ratio) is shown in Figure 6. It can be found that the equal-effect variation is positively related to the number of grooves and increases with the increasing in the number of grooves. The different colors in the simulation results represent the strain values in different ranges. The strain layer in reverse extrusion is divided into two layers. The first layer is the inner wall of the cylindrical part close to the working zone of the punch, and the strain value is between 1.88–3.75 mm/mm; the strain values of other parts are small and distributed between 0–1.88 mm/mm. With the increase in the number of grooves, the strain stratification in RE cylindrical parts becomes more and more obvious, and the range of large strain layers is also increasing. From the cylinder wall, the strain influence ranging from the inner wall to the outer wall gradually expands. Due to the addition of shear structure and rotation factors during RE process, the metal at the bottom of the punch has a large shear deformation.

Therefore, compared with reverse extrusion, RE can bring greater deformation strain. The deformation dead zone under the original backward extrusion punch is improved. The metal under the punch obtains large strain during deformation through the groove structure and flows to the cylinder wall through backward extrusion, which expands the range of the strain layer of the cylinder wall and improves the overall deformation strain. The groove structure with the number of 8 has the widest strain range and the highest cylinder strain value.

The histogram of simulated strain-value distribution of cylindrical parts is shown in Figure 7. The average equivalent strain of back extrusion is 1.31 mm/mm, and the average equivalent strain increases with the increase in the number of grooves, up to 3.95 mm/mm of eight grooves, an increase of 201.5%. The proportion of strain values in different intervals in the histogram intuitively explains the increase in strain values. The abscissa strain values range from 0–15 mm/mm, each interval is 0.75 mm/mm, and the ordinate is the percentage of each strain-value interval. From the abscissa, the interval value occupied by the strain value is increasing, indicating that the maximum strain value increases gradually with the increase of the number of grooves. From the vertical coordinate, the percentage of the range of low strain values gradually decreases with the increase in the number of grooves, indicating that the range of low strain values in cylindrical parts is gradually decreasing while the range of high strain values is gradually expanding. That is to say, more metals take part in the deformation, and their own deformation is also increasing.

In the postprocessing of the simulation, the blank model is set with five marking points. According to the flow trace points of the marking points, the metal flow characteristics in the deformation process of reverse extrusion and RE are analyzed and determined. The 2D flow path of the five marked points under different groove structures is shown in Figure 8. This group of figures is the X-Y plane. P1–P5 points are distributed inward from the punch edge (x = 0, the radius direction is the Y direction). The distance from the deformed point to the center along the radius direction should be the diagonal distance of X-Y. The reverse extrusion metal first undergoes compression deformation and then flows out along the punch. It can be seen from Figure 8a that only point P5 flows to the cylinder wall, while point P1–P4 is still below the punch. The farthest point moves less than 0.25 mm along the X direction, which fully reflects the dead-zone characteristics of reverse extrusion deformation. With the increase in the number of grooves, the movement of each marking point along the X direction increases obviously. Points P1 and P2 have flowed to the cylinder wall, and point P3 has reached the edge of the punch working zone. The metal flow path is no longer a nearly linear flow, but with a certain circular motion, which also shows that the metal can increase the deformation stroke and increase the strain under the groove structure.

The comparison of forming load, strain distribution and metal flow shows that the increase in the number of grooves is beneficial to deformation. According to the FEM results of 0, 2, 4, 6 and 8 groove gradients, the optimal number of grooves can be determined as eight grooves.

In addition, the number of grooves is determined as eight, and the influence of groove depth on deformation is also worth exploring. It is inferred from the common sense that if the groove depth is shallow, the degree of shear deformation will decrease, and if the groove depth is deep, forming defects such as folding and tearing may occur. The following will analyze and determine the optimal groove depth through the gradient simulation of groove depth of 5 mm, 6 mm, 7 mm and 8 mm. The groove depth of the eight-groove structure above is 5 mm, so only 6–8 mm is analyzed and compared below.

The variation of the simulated forming load and torque under different groove-depth gradients is shown in Figure 9. The forming load shows a stable trend after the stroke of 60 mm, and the average value of the load after 60 mm is taken as the final comparison value. The forming loads with groove depth of 6 mm, 7 mm and 8 mm are ~3970, ~3850 and ~3740 kN, respectively, and the torque is ~5.3 × 10^4^, ~5.8 × 10^4^, ~6.3 × 10^4^ N·m, respectively. It can be seen from Figure 5 that the load and torque under the groove depth of 5 mm is ~412 kN and ~4.8 × 104 N·m.

The load corresponding to each groove depth is close, but the overall trend remains the same. With the increase in groove depth, the load decreases and the torque increases. It shows that on the basis of increasing the number of grooves, deepening the groove depth can continue to reduce the forming load, and the reduction in forming load is accompanied by the continuous increase in torque.

The groove depth has a direct impact on the strain value and distribution. The simulated strain distribution results under the groove-depth gradient are shown in Figure 10. Figure 10a–c shows the simulated strain distribution of cylindrical parts. The effect of groove depth on strain is mainly reflected in two aspects. On the one hand, from the point indicated by the red dotted circle at the bottom of the cylinder, the strain color changes from dark blue (1.88–3.75 mm/mm) to cyan (5.63–7.50 mm/mm), and the strain value at the bottom of the cylinder is constantly increasing. On the other hand, from the maximum diameter of the punch indicated by the white arrow, the red strain value (13.1–15.0 mm/mm) is expanding inward from the maximum diameter of the punch. Figure 10d–f shows the percentage of each strain interval. The strain of 0–15 mm/mm is divided into five intervals, and the percentage of each interval is marked in the figure. The percentage of the 0–3 mm/mm section decreases from 56.38% to 45.53%, the percentage of 12–15 mm/mm section increases from 4.26% to 7.63%, while the percentage of the groove depth of 7 mm and 8 mm in the 3–12 mm/mm section increases significantly compared with 6 mm, and there was little difference between 7 mm and 8 mm. The average strain increases from 4.46 mm/mm to 5.43 mm/mm, the average depth increases by 1 mm, and the strain increases by 0.5 mm/mm. The above data show that the metal deformation at the bottom of the punch increases with the increase in groove depth, and the range of high strain inward from the maximum diameter of the punch expands. For the whole cylindrical part, the increase in the medium strain interval (3–12 mm/mm) is not obvious after the groove depth is 7 mm, while the low strain interval (0–3 mm/mm) and high strain interval (12–15 mm/mm) change greatly under the influence of the groove depth. The decrease in the percentage of the low strain interval and the increase in the percentage of the high strain interval are the main reasons for the increase in the strain with the increase in the groove depth.

From the point of view of load, the increase in groove depth will continue to reduce the forming load; in terms of strain distribution, the increase in groove depth will continue to increase the average equivalent strain, but the deeper groove depth does not mean it is better. The restriction on the groove depth lies in the metal flow. The increase in groove depth will increase the movement speed and inflow and outflow speed of metal along the circumferential direction. If the speed exceeds the appropriate value, it will have a negative impact on the forming; that is, the generation of folding defects. Using the flow example in Figure 11, there is a velocity difference between the inflow area and outflow area in the groove along the Y and Z directions, and the folding defect also comes from this. As shown in Figure 12, the simulation results show that when the groove depth is 7 mm and 8 mm, folding occurs at the groove opening, and the formed folding is spirally distributed along the inner wall of the cylindrical part. The folding defect is mainly caused by the mismatch of the absolute difference of velocity in all directions caused by the deep groove. As shown in Table 2, the velocity values of the marked points in the left and right areas in the Y and Z directions are averaged and the absolute value of the velocity average makes a difference. The absolute difference of the velocity average in the X direction is about 1 mm/s, the absolute difference of the velocity average in the Y direction increases from 1.45 mm/s to 4.21 mm/s, and the absolute difference of the velocity average in the Z direction increases from 2.85 mm/s to 4.22 mm/s. Combining the absolute difference of the average velocity in each direction with the folding defects shown in the simulation results, it shows that folding defects will occur when the absolute difference of the average velocity in Y and Z directions exceeds 4.0 mm/s. The velocity difference in the X and Y directions will cause the appearance of the covering layer in the process of the groove pushing the metal to move along the circumferential direction. The left area of the groove belongs to the passive flow area, and the right area belongs to the active flow area. Too fast a velocity value of the active flow area will cause the formation of the covering layer in the left area of the groove. The velocity difference in the Z direction will cause voids in the groove. The metal flow in the inflow area is mainly affected by the axial extrusion velocity due to the large depression and pushing velocity of the groove in the outflow area. Therefore, the Z velocity value is small. When the outflow volume is large and the inflow volume is insufficient, voids will appear in the left area of the groove, providing conditions for the emergence of the covering layer. Under the allowable conditions, the groove depths of 5 mm and 6 mm do not fold. Compared with the two groups of Z-direction velocity values, higher values can make the metal quickly conduct up and down shear deformation, thus providing high strain. Therefore, the groove depth of 6 mm is the best choice.

It can be seen from Figure 8 that when the angle of the flaring angle is 3°, the extrusion load is ~3970 kN and the torque is ~5.3 × 10^4^ N·m; the change of extrusion load and torque after the increase of flaring angle is shown in Figure 13. The extrusion load is concentrated at about ~4000 kN, with little difference, and the torque is about ~6.1 × 10^4^ N·m. According to the comparison, the increase in flaring angle has little effect on extrusion load and torque in the range of 3–11°.

The average value of the simulated equivalent strain with the flaring angle of 3° is 4.46 mm/mm. The histogram of the equivalent strain distribution of the cylindrical part after increasing the flaring angle is shown in Figure 14. It can be found that the gradient value of the strain interval is changed to 3.6 mm/mm. The average values of equivalent strain from 5–11° are 4.92 mm/mm, 5.04 mm/mm, 5.02 mm/mm and 4.96 mm/mm, respectively. With the increase in angle, the strain first increases and then decreases, and the strains at 7° and 9° are very close. From the percentage of strain interval, the percentage of the 3.6–14.4 mm/mm interval changes little, indicating that changing the flaring angle has little effect on the interval of the medium-strain layer, which is mainly affected by the number of grooves (area ratio) and the depth of grooves. The percentage of the low-strain layer (0–3.6 mm/mm) decreases while that of the high-strain layer (14.4–18 mm/mm) increases. This change indicates that the change of the flaring angle can promote the metal at the bottom of the punch to flow rapidly to the cylinder wall, although the groove structure at the bottom of the punch can bring considerable shear strain to the metal, From the forming point of view, it is also the key that the metal participating in the deformation can quickly flow out to the cylinder wall, so that the bottom metal can participate in the deformation. Therefore, there is a downward trend in the average value of equivalent effect after the flaring angle is greater than 9°. This trend just shows the existence of this phenomenon.

With the increase in the flaring angle, the average equivalent strain of the cylindrical part increases first and then decreases. In order to find out the reason for this change, the blank is marked and its flow path is tracked. As shown in Figure 15, three rows of marking points are set in the blank before extrusion. The marking number of five points in each row increases from the inside to the outside. The positions of the points after deformation are uniformly recorded and compared. The red line in Figure 15b is the initial position. It can be seen from the comparison that the position of point P4 has changed significantly. When the flaring angle is 3°, the included angle between the final position and the initial position is ~90°, and when the flaring angle is 11°, the included angle is ~60°. With the increase of the flaring angle, the included angle between the final position and the initial position decreases, which indicates that point P4 flows to the cylinder wall earlier. The included angle of point P3 is ~179° at 3°, which is ~10 mm away from the edge of the inner wall. When the angle of the flaring angle increases to 11°, the included angle of point P3 decreases to ~165° and has flowed to the edge of the inner wall. The position comparison of point P8 in the second row before and after deformation is similar to that of P4 and P3. According to the comparison of point-tracking results, the change of the cone apex angle has a great impact on the metal flow under the punch from inside to outside (along the diameter direction). The increase of flaring angle will accelerate the metal flow under the punch to the cylinder wall. It is not difficult to see from the distribution position of marked points in the figure that the groove structure promotes the metal flow along the circumference under the action of rotation. When flowing under the punch, it will experience groove shear to increase the strain, so flowing too fast out of the shear deformation zone will reduce the strain.

From the change of forming load and torque, the change of the flaring angle has little effect on it. From the strain distribution, the average equivalent strain will decrease when the flaring angle is greater than 9°. After the point-tracking study of the simulation results, it is found that the increase in the flaring angle will shorten the time for the metal at the bottom of the punch to flow to the cylinder wall, and accelerate the flow of the metal that has participated in the deformation to the cylinder wall. Extrusion forming belongs to equal volume forming, so the outflow of metal will be accompanied by the inflow of metal. The outflow of metal involved in shear deformation as soon as possible will make the metal not involved or indirectly involved in the bottom flow into the groove at the bottom of the punch for shear deformation, which can improve the overall deformation strain and deformation uniformity of the cylindrical part. Combined with the distribution of equivalent strain and the law of metal flow, the angle range of the flaring angle can be 7°–9°.

### 3.2. Microstructure of Mg-Gd-Y-Zn-Zr Alloy

Figure 16 exhibits the die and object of Mg-Gd-Y-Zn-Zr alloy after RE process on the basis of FEM results. The billet and dies are heated at 420 °C and 450 °C for 3 h, respectively, in order to prevent the heat loss during the assembly process. The axial speed is 1 mm/s and the rotation speed is 3 r/min after the sizing strip of the punch completely entering the blank until the RE achieves.

Figure 17 and Figure 18 shows the microstructure of initial experimental and REed Mg-9Gd-4Y-2Zn-0.5Zr (wt.%) alloy, respectively. The initial alloy is homogenized at 520 °C for 16 h and pre-deformed by RUE method for 1 pass at 420 °C with the extrusion ratio of 1.98 [32]. It can be found that the experimental alloy consists of Mg matrix, lamellar LPSO phases, block-shaped LPSO phases, Mg_5_RE phases and cubic RE-riched phases in our previous study [33]. The initial alloy is mainly composed of coarse deformed grains and fine equiaxed grains, which form an obvious bimodal structure. The coarse deformed grains are mostly elongated along ED, and the fine grains are distributed around the coarse grains randomly. The occurrence of a bimodal microstructure is attributed to the generation of dynamic recrystallization (DRX) behavior. However, the ability of refining grains by DRX behavior is limited, which brings about low proportion of DRXed grains.

In addition, the microstructure of Mg-Gd-Y-Zn-Zr alloy in the inner, middle and outer regions after the RE process are also investigated, as shown in Figure 18. The thickness of the cylinder wall is approximately 30 mm; thus, in order to explore the trend from inner region to outer region, 10 mm of observation surface in different regions are selected. It is obvious to find that the cylinder wall from the inner region to outer region exhibits a gradient microstructure. The degree of grain breakage becomes more and more obvious; the grain size in the inner region is the smallest, followed by the middle, and the grain size in the outer region is the largest. Moreover, the coarse grain and block-shaped LPSO phases gradually decrease, ranging from the outer region to the inner region owing to the different metal flow and strain during the RE process, which is consistent with the FEM results. The specific results will be shown in our further study.

There are two limitations in this paper. First, our friction coefficient is a constant value of 0.3, which depends on the lubricant actually used. However, deformation is a dynamic process, and the friction coefficient should be dynamic [34]. Second, the simulation process is set to isothermal deformation; even if we have a heat-preservation device, there will be heat loss during the actual deformation process [35].

## 4. Conclusions

Finite element simulation of a punch structure and its effect on the microstructure evolution of Mg-Gd-Y-Zn-Zr alloy via RE method were investigated in the paper. The structure with a groove number of 8 could reduce the forming load of RE by 28.35% compared with reverse extrusion, and increase the average equivalent strain by 201.5%. The increase in the groove depth would produce a difference in the flow velocity of the metal in the groove. The metal in the left and right areas of the groove could fold when the velocity difference along the Y and Z directions exceeded 4 mm/s, so the appropriate value of the groove depth was 6 mm. The influence of the flaring angle on the deformation was mainly to shorten the time for the metal at the bottom of the punch to flow to the cylinder wall, so as to expand the deformation range and forming uniformity. The most suitable flaring angle was 7–9°. The cylinder parts could be formed with good quality according to the optimized FEM results. The cylinder wall from the inner region to the outer region exhibited a gradient microstructure owing to the different metal flow and strain during the RE process. The effect of RE on the mechanical properties of Mg alloys at different deformation rates will be regarded as further study.

## Figures and Tables

**Figure 1 materials-15-05248-f001:**
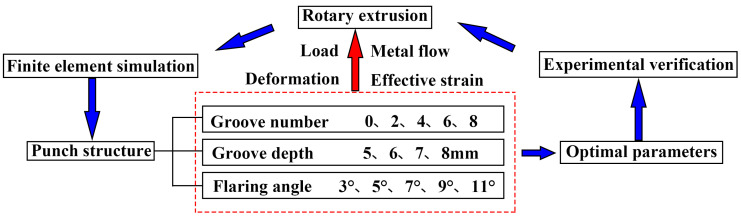
The research flow of rotary extrusion deformation in this paper.

**Figure 2 materials-15-05248-f002:**
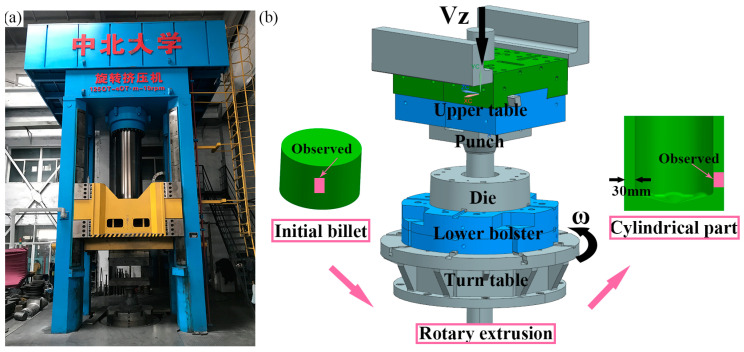
(**a**) Rotary extrusion (RE) forming experimental equipment and (**b**) experimental die.

**Figure 3 materials-15-05248-f003:**
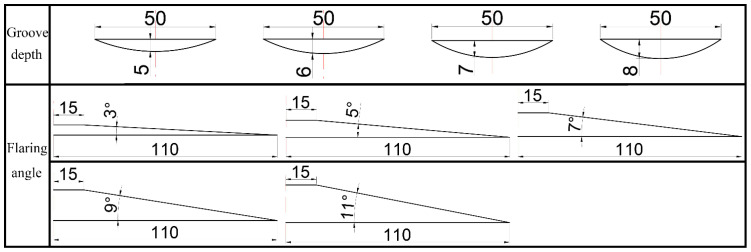
Schematic diagram of groove depth and flaring angle.

**Figure 4 materials-15-05248-f004:**
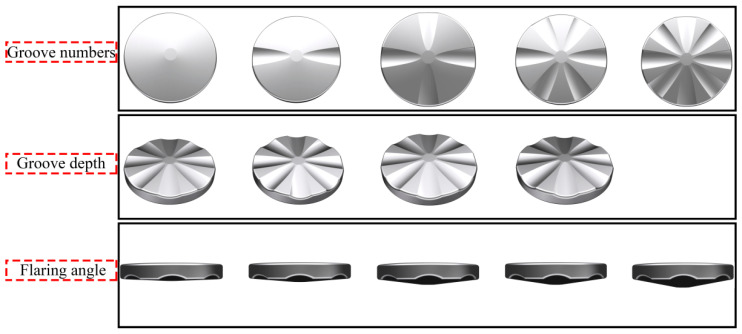
Schematic diagram of finite element simulation punch models.

**Figure 5 materials-15-05248-f005:**
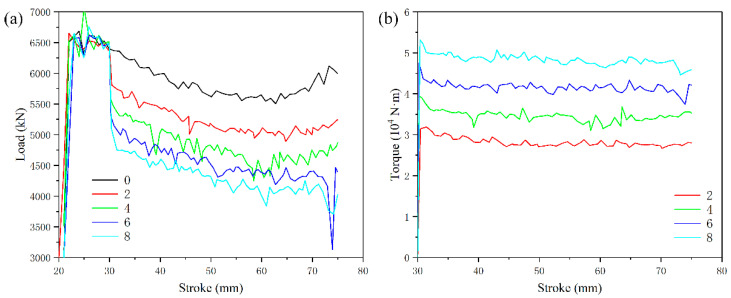
Variation of extrusion load and torque of cylindrical parts under gradient of groove numbers: (**a**) load; (**b**) torque.

**Figure 6 materials-15-05248-f006:**
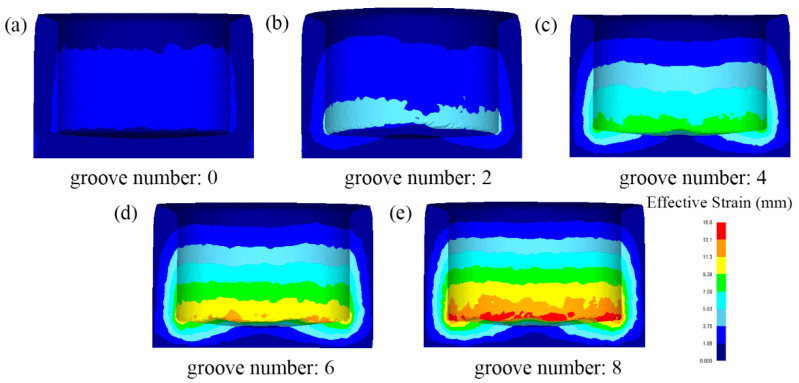
Equivalent strain distribution diagram of cylindrical parts under gradient of groove numbers, (**a**) 0, (**b**) 2, (**c**) 4, (**d**) 6, (**e**) 8.

**Figure 7 materials-15-05248-f007:**
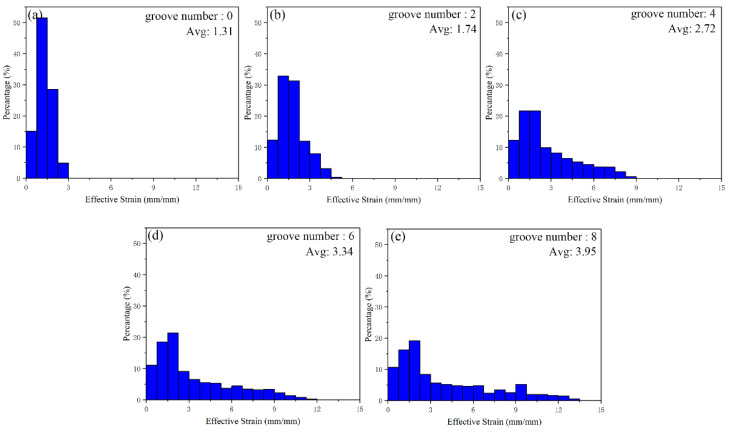
Equivalent strain histogram of cylindrical parts under gradient of groove numbers, (**a**) 0, (**b**) 2, (**c**) 4, (**d**) 6, (**e**) 8.

**Figure 8 materials-15-05248-f008:**
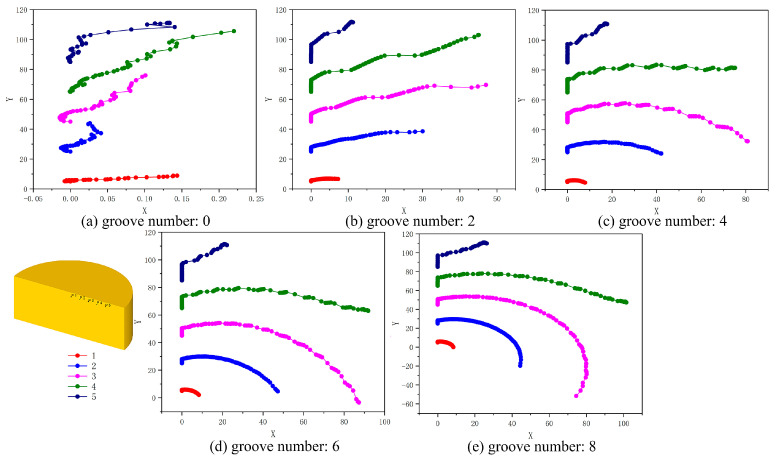
Two-dimensional diagram of flow track of marked points in cylindrical parts under gradient of groove numbers, (**a**) 0, (**b**) 2, (**c**) 4, (**d**) 6, (**e**) 8.

**Figure 9 materials-15-05248-f009:**
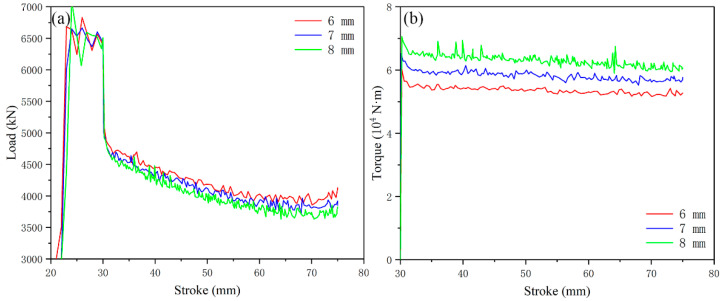
Influence of different groove depths on extrusion load and torque: (**a**) load; (**b**) torque.

**Figure 10 materials-15-05248-f010:**
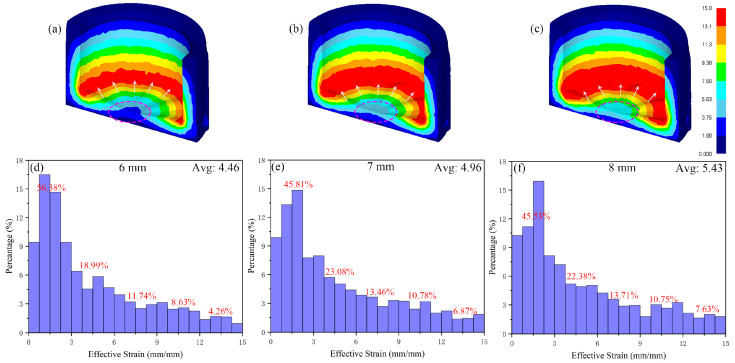
Strain distribution (**a**–**c**) and histogram of simulated strain distribution (**d**–**f**) at different groove depths.

**Figure 11 materials-15-05248-f011:**
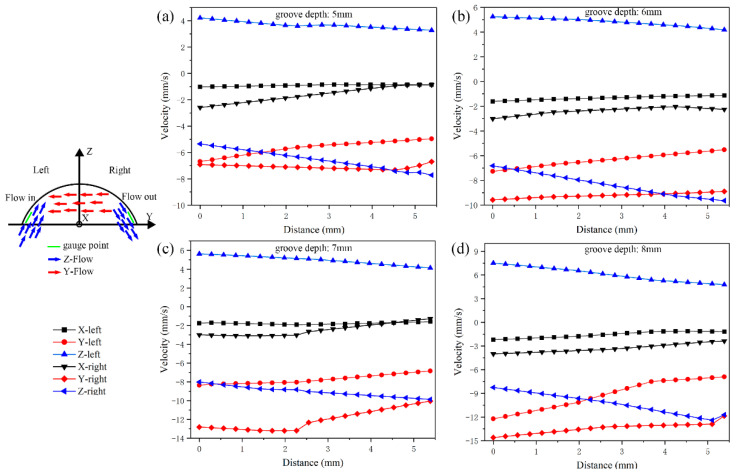
Velocity distribution diagram of marked points in left (inflow) and right (outflow) areas of different groove depths, (**a**) groove depth: 5 mm, (**b**) groove depth: 6 mm, (**c**) groove depth: 7 mm, (**d**) groove depth: 8 mm.

**Figure 12 materials-15-05248-f012:**
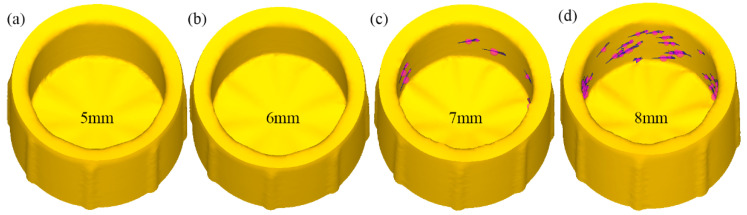
Simulated folding distribution map of different groove depths, (**a**) groove depth: 5 mm, (**b**) groove depth: 6 mm, (**c**) groove depth: 7 mm, (**d**) groove depth: 8 mm.

**Figure 13 materials-15-05248-f013:**
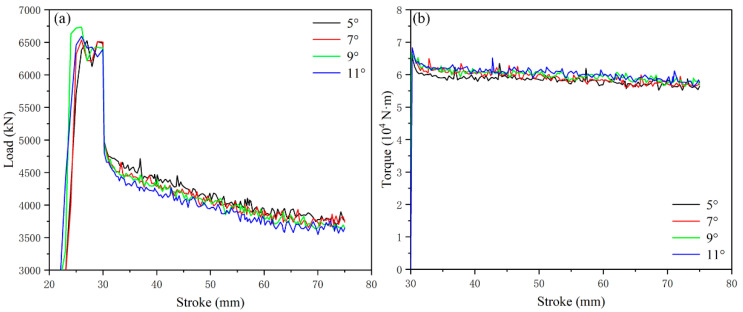
Influence of flaring angle on extrusion load and torque: (**a**) load, (**b**) torque.

**Figure 14 materials-15-05248-f014:**
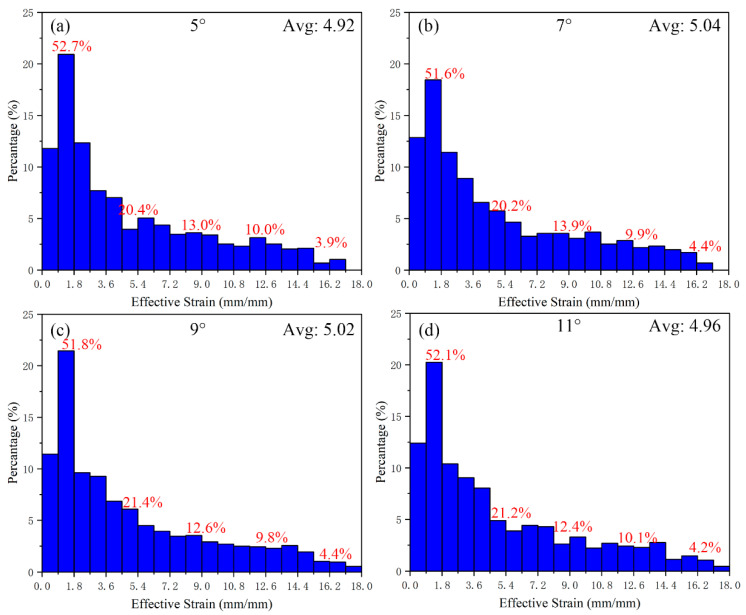
The equivalent strain distribution of cylindrical parts corresponds to the flaring angle gradient, (**a**) flaring angle: 5°, (**b**) flaring angle: 7°, (**c**) flaring angle: 9°, (**d**) flaring angle: 11°.

**Figure 15 materials-15-05248-f015:**
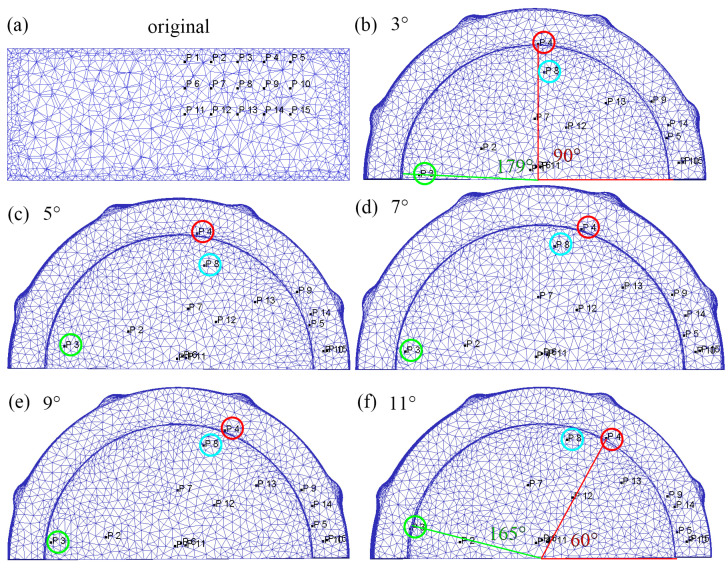
The flow distribution of the marked point of the cylindrical parts corresponding to the gradient of the flaring angle, (**a**) original, (**b**) flaring angle: 3°, (**c**) flaring angle: 5°, (**d**) flaring angle: 7°, (**e**) flaring angle: 9°, (**f**) flaring angle: 11°.

**Figure 16 materials-15-05248-f016:**
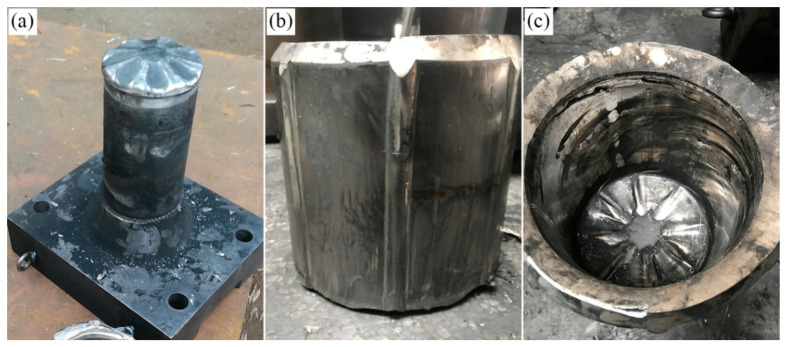
The die and object of Mg-Gd-Y-Zn-Zr alloy after BE process, (**a**) punch, (**b**) die, (**c**) object.

**Figure 17 materials-15-05248-f017:**
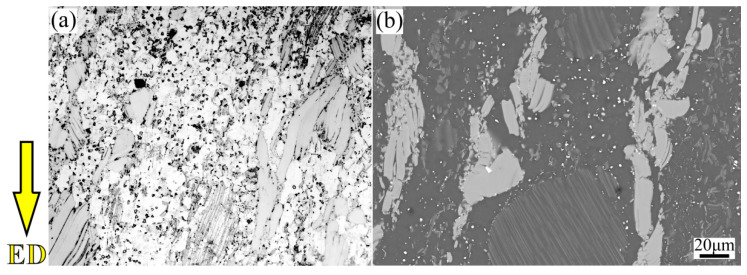
The OM and SEM images of the initial Mg-Gd-Y-Zn-Zr alloy, (**a**) OM, (**b**) SEM.

**Figure 18 materials-15-05248-f018:**
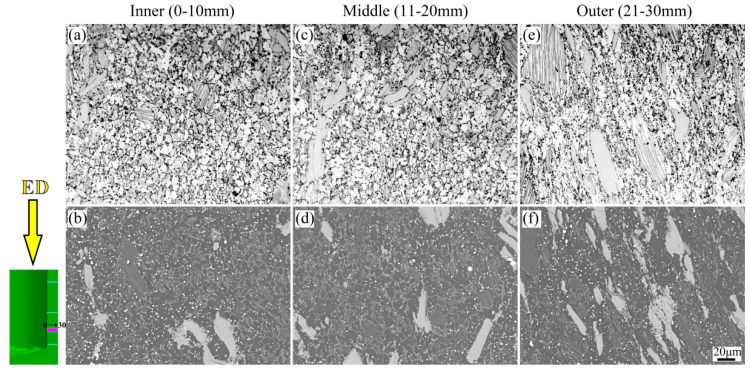
The OM and SEM images from inner region to outer region of Mg-Gd-Y-Zn-Zr alloy via RE method, (**a**) OM: Inner, (**b**) SEM: Inner, (**c**) OM: Middle, (**d**) SEM: Middle, (**e**) OM: Outer, (**f**) SEM: Outer.

**Table 1 materials-15-05248-t001:** Finite element simulation gradient comparison table.

Grooves	Gradient Values					
Number	0	2	4	6	8	Depth: 5 mm flaring angle: 3°
Area ratio	0%	16.12%	33.30%	48.39%	64.49%	
Depth	5 mm	6 mm	7 mm	8 mm		Number: 8 flaring angle: 3°
Flaring angle	3°	5°	7°	9°	11°	Umber: 8 Depth: 7 mm

**Table 2 materials-15-05248-t002:** The mean value and the corresponding difference of the marking point velocity.

Depth	Region	$\bar{X}$ (mm/s)	Absolute Difference (mm/s)	$\bar{Y}$ (mm/s)	Absolute Difference (mm/s)	$\bar{Z}$ (mm/s)	Absolute Difference (mm/s)
5 mm	left	−0.91	0.72	−5.64	1.45	3.68	2.85
	right	−1.63		−7.09		−6.53	
6 mm	left	−1.33	1.05	−6.35	2.87	4.82	3.51
	right	−2.38		−9.22		−8.33	
7 mm	left	−1.77	0.68	−7.93	4.07	4.98	4.05
	right	−2.45		−12.0		−9.03	
8 mm	left	−1.56	1.73	−9.19	4.21	6.08	4.22
	right	−3.29		−13.4		−10.3	

## Data Availability

The data presented in this study are available on request from the corresponding author.

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
