# Peer review of "Finite Element Simulation of Punch Structure and Its Effect on Microstructure Evolution of Mg-Gd-Y-Zn-Zr Alloy via Rotary Extrusion Method"

_materials, 2022, doi:10.3390/ma15155248_

Round 1

Reviewer 1 Report

The numerical simulation is the core of this work. Nevertheless, the setting and the conditions are not described. The doubts on the model setting are furthermore increased by the results of the simulations in which noise (fig. 4) and non-symmetric distributions (fig. 5) are visible. This could result from a statistical distribution of properties and inhomogeneities, or, most probably, it is due to a bad mashing and a bad marching method of the model. In the referee’s opinion, this makes the results non-trustable.

The text is poorly written, the figures included in Materials and Methods with their captions lack in resolution and details, and the sections contents are unbalanced.

Author Response

Thank you for your comments concerning our manuscript entitled “Finite element simulation of punch structure and its effect on microstructure evolution of Mg-Gd-Y-Zn-Zr alloy via rotary extrusion method”. Those comments are all valuable and very helpful for revising and improving our paper, as well as the important guidance to our research. We have studied the comments carefully and have made corrections which we hope meet with approval.

In the Materials and Methods section, We have supplemented the setting conditions and details of numerical simulation. The original manuscript had load fluctuations, which belong to the normal range as shown in Fig. 4. The deformation mode is complex, including circumferential rotary shear deformation and axial extrusion deformation, so the mesh distortion phenomenon will appear in the simulation process during the RE process. In view of this phenomenon, we use mesh rezoning and volume compensation in Deform-3D software to reduce the calculation error. The uneven strain distribution in Fig. 5 in the original manuscript is a feature of RE deformation, because with the increase of rotation revolutions, the number of shear deformation will also increase, so the phenomenon of strain accumulation appears. We have made corresponding supplements to the content of the article, which can be seen in the revised manuscript.

Reviewer 2 Report

The present research explores the effect of punch geometry likes the number of grooves, the groove depth, and flaring angle on strain distribution and its effect on microstructure evolution. The finite element simulation software was employed to simulate the effect of punch structure on rotary extrusion of Mg-Gd-Y-Zn-Zr alloy. It can be accepted after addressing the following issues:

1- Page 1, Introduction section, third line: the "et al" must be changed to "etc.".          

2- Page 2, Second paragraph: 7th and 8th lines: the "lo" and "as-castes" must be modified.

3- It is necessary to provide a schematic for specimen and deformed one in the Materials and Methods section, if it is possible.

4- The input mechanical properties of the alloy and any useful information which were used in the simulation, must be specified in a table in the Materials and Methods section.

5- It is recommended that the different components of the equipment are introduced in Fig. 1 (b).

6- There is no information about preparation method for example polishing and etching in the Materials and Methods section. Furthermore, the location of sampling should be specified.

7- Page 5, the last line of first paragraph: Each torque value should be assigned to a groove number because the word "respectively" was used in the sentence.

8- Page 5, third line of second paragraph: "…by relevant researchers": The references must be mentioned.

9- Page 5, the last line of third paragraph: There is no plastic deformation in the dead metal zones (DMZ), i.e. the strain field is zero in the DMZ. How this discrepancy can be explained?

10- Page 8: I could not find Figure 3-3 and Figure 3-10.

11- Page 8: the first line of second paragraph: "the groove depth of the groove" should be changed to "the groove depth".

12- Page 13: section 3-2: The simulation results were not verified by the experimental observations. It needs more discussions.

13- Page 14, the second line of first paragraph: It is not clear that how bimodal microstructure can be attributed to the dynamic recrystallization? The coarse grains elongated along ED direction (Fig. 16) and they are not recrystallized grains. 

Author Response

Thank you for your comments concerning our manuscript entitled “Finite element simulation of punch structure and its effect on microstructure evolution of Mg-Gd-Y-Zn-Zr alloy via rotary extrusion method”. Those comments are all valuable and very helpful for revising and improving our paper, as well as the important guidance to our research. We have studied the comments carefully and have made corrections which we hope meet with approval. The changes are highlighted in yellow.

  1. The "et al" has beenchanged to "etc.", which can be seen in the revised manuscript.
  2. The spelling errors have been corrected in the revised manuscript.
  3. It is modified that the deformation schematic is added in Figure 2 (b), which can be seen in the revised manuscript.
  4. Information about the simulation setup is supplemented in the Materials and Methods section, which can be seen in the revised manuscript.
  5. The different components of the equipment are introduced in Fig. 2(b), which can be seen in the revised manuscript.
  6. The information about the preparation method, such as polishing and etching, has been supplemented in the Materials and Methods section, and the sampling location is indicated in Figure 2 (b). These modifications can be seen in the revised manuscript.
  7. The each torque valuehas been assigned to a groove number in the revised manuscript.
  8. The RE deformation can effectively reduce the forming load has been confirmed by relevant researchers[1,2].

[1].X. Ma, M.R. Barnett, Y.H. Kim, Experimental and theoretical investigation of compression of a cylinder using a rotating platen, Int. J. Mech. Sci. 45 (2003) 1717-1737.

[2].X. Ma, M.R. Barnett, Y.H. Kim, Forward extrusion through steadily rotating conical dies. Part II: theoretical analysis, Int. J. Mech. Sci. 46 (2004) 465-489.

  1. The characteristics of traditional reverse extrusion deformation are that the metal at the bottom of the punch is mainly relying on compression deformation. The strain is low, and the metal fluidity is not good during the traditional reverse extrusion process. Due to the addition of shear structure and rotation factors during the RE process, the metal at the bottom of the punch has a large shear deformation and can flow fully. This is the difference between the above two deformation methods. The deformation dead zone mentioned in the original paper does not mean that the strain value in this area is 0, but the strain value is small, which is generally called the difficult deformation zone. It has been corrected in the revised manuscript.
  2. The writing errors have been corrected in the revised manuscript.
  3. The writing errors have been corrected in the revised manuscript.
  4. The optimized punch structure is verified by the experiment in section 3.2. The cylindrical parts were formed via the RE method, and the microstructure of the cylindrical parts was also observed. There is a strain gradient in the cylinder wall in the FEM results, and the purpose of simulation optimization is to adjust the punch structure to increase the strain value of the cylinder wall. The magnitude of the strain value is positively correlated with the deformation degree, and the deformation degree is positively correlated with grain refinement and phase breakage. Therefore, the simulation optimization results by observing the microstructure distribution of the cylindrical parts. As shown in Fig. 17and Fig. 18, compared with the initial microstructure, the degree of grain refinement and second phase fragmentation in the range of 0-20 mm of the barrel wall is larger, while the range of 21-30 mm is similar to the initial microstructure. It indicates that the degree of RE deformation is large, and the range is wide.
  5. The initial alloy is obtained by RUE deformation after 1 pass. During the deformation process, the amount of deformation is limited, and  insufficient deformation leads to low DRX proportion. The coarse grains in the initial alloy are deformed grains, while the small equiaxed grains are formed by DRX behavior during the deformation process, and the bimodal structure is also formed.

Reviewer 3 Report

1.      Uppercase and lowercase of the title and subsection of the present manuscript should be revised.

2.      The way to type the affiliation and emails of the authors is not proper.

3.      Qualitative results need to be included in the abstract section rather than only qualitative results.

4.      Please, reorder keywords based on alphabetic order.

5.      What is the novelty of the present manuscript? I do not see something really new in this article. The finite element study of microstructure Mg-Gd-Y-Zn-Zr alloy has been widely studied. Also, a similar article has been published by the authors as follows: Effect of rotating shear extrusion on the microstructure, texture evolution, and mechanical properties of Mg-Gd-Y-Zn-Zr alloy. Journal of Alloys and Compounds. Volume 906, 15 June 2022. https://doi.org/10.1016/j.jallcom.2022.164406 makes the present research lack novelty. The authors were given chance to address this issue.

6.      Research flow needs to be explained in the form of an illustrative figure in the materials and methods section.

7.      The authors need to explain the advantage of adopting finite element simulation to solve engineering problems such as low cost, minimizing the time required, and others. Please include this important point in the introduction and/or conclusion section. Also, to support this explanation the authors need to adopt suggested references published by Materials, MDPI as follows: Tresca Stress Simulation of Metal-on-Metal Total Hip Arthroplasty during Normal Walking Activity. Materials (Basel). 2021, 14, 7554. https://doi.org/10.3390/ma14247554

8.      Microstructure evaluation is very minimal information. More detail is needed in this section.

9.      The authors should be giving detailed information regarding tools that there used to clearer information to the reader.

10.   Accuracy and tolerance are needed to be given due to discussion regarding different results in the further studies with different tools used.

11.   The finite element model with its boundary condition needs to be described.

12.   A Mesh convergence study should be described.

13.   The meshing strategy for model discretization is missing.

14.   Detailed information on element type, number of nodes, and number of elements should be given.

15.   In Figure 16 and Figure 17, since the figure scale is the same, just make it in one scale for all of the divided figures.

16.   Comparation on the results from other literature with similar content should be done by authors.

17.   The limitation of the present study needs to be stated before the conclusion section.

18.   Please rewrite the conclusion section, the present form is not solid. Also make it into paragraphs, not point-by-point as present form.

19.   Further study needs to be explained in the conclusion section.

20.   Please recheck the English’s used and enhance the language style.

21.   Make sure the template is already proper based on materials guidelines.

Author Response

Thank you for your comments concerning our manuscript entitled “Finite element simulation of punch structure and its effect on microstructure evolution of Mg-Gd-Y-Zn-Zr alloy via rotary extrusion method”. Those comments are all valuable and very helpful for revising and improving our paper, as well as the important guidance to our research. We have studied the comments carefully and have made corrections which we hope meet with approval. The changes are highlighted in green.

  1. The uppercase and lowercase of the title and subsection of the present manuscripthave been revised, which can be seen in the revised manuscript.
  2. The affiliation and emails of the authorshave been corrected in the revised manuscript.
  3. The qualitative results have been added in the abstract section, which can be seen in the revised manuscript.
  4. The keywords have been corrected in the revised manuscript.
  5. Rotary extrusion (RE) is a severe plastic deformation (SPD) technology with application prospects. The shear structure of punch is very important for RE. This paper mainly studies the effect of groove shear structural parameters (groove number, groove depth, flaring angle) on the RE method. The results show that the effect of the groove number on the deformation strain of the alloy is increased by increasing the number of shear deformation. The change of groove depth will lead to metal flow velocity difference during the RE process, and the groove depth to avoid folding defects is given. The change of flaring angle can change the velocity of metal flowing from the bottom of the punch to the cylinder wall, thus affecting the deformation strain of the alloy. These three research results show the influence law of groove structure on the RE process, and also provide a reference for the follow-up study of RE. The reviewer said that these research contents were not mentioned in our previouslypublished articles.
  6. The research flow has been added to the revised manuscript, as shown in Fig. 1.
  7. The advantages of the Finite element simulation are illustrated in the last paragraph of the introduction section, and [1] is cited as supporting literature.

[1]M.I. Ammarullah, I.Y. Afif, M.I. Maula, T.I. Winarni, M. Tauviqirrahman, I. Akbar, H. Basri, E. van der Heide, J. Jamari, Tresca Stress Simulation of Metal-on-Metal Total Hip Arthroplasty during Normal Walking Activity. Materials. 24 (2021)14:7554.

  1. The shearing structure of the punch is very important to the RE process, which directly affects the strain and formability of RE. It is the premise of industrial production to increase the proportion of fine grains by increasing the strain to improve the properties of Mg alloys, and to ensure that there are no folding defects in the forming process of components. This paper focuses on the influence of groove number, groove depth and flaring angle on RE of the groove shear structure. The detailed microstructure analysis has been published as follows: Effect of rotating shear extrusion on the microstructure, texture evolution, and mechanical properties of Mg-Gd-Y-Zn-Zr alloy. Journal of Alloys and Compounds. Volume 906, 15 June 2022. The analysis of punch structure and microstructure evolution are two parts of research that do not overlap. Subsequent new research contents of RE will continue to be published.
  2. The detailed informationabout toolshas been supplemented in the Microstructure Characterization section (2.2), which can be seen in the revised manuscript.
  3. The results were obtained by Deform-3D software, and the specific values were provided by it.

11、12、13、14. These problems (11, 12, 13, 14) belong to the category of finite element simulation (FEM), and we have made a unified supplement in the materials and methods.

  1. The figure scale in Fig.16 and Fig.17 of the original manuscript has been corrected in the revised manuscript.
  2. The research content of this paper is the optimization of punch groove structure and the influence of groove structure parameters on the RE process. And it is aimed at the forming of large cylindrical parts, not the preparation of small components. The research on the structure of this size REgroove has not been reported. At the same time, we pay more attention to optimizing the shear structure designed by ourselves.
  3. The limitation of the present study have been stated before the conclusion section, which can be seen in the revised manuscript.
  4. The conclusion section has been rewritten in the revised manuscript.
  5. The further study has been explained in the conclusion section, which can be seen in the revised manuscript.

20、21. These problems have been corrected in the revised manuscript.

Round 2

Reviewer 1 Report

The manuscript has not been improved sufficiently. Materials, boundary conditions, and contact interactions were not specified. The simulation should be based on an explicit transient model: which time-step duration and the simulation duration have been set? At which time-step the authors picked the results reported?

The issues detected regarding text and materials and methods (Figures 3 and 4) are still included in the document.

Author Response

1.Thank you for your comment. The material model, boundary conditions and friction contact settings are described in Section 2.1 of the article. In the simulation process, the step length is 1 mm/step, and the total stroke is 75mm. The final simulation result with the stroke of 75 mm is selected as the analysis data. We have made corresponding supplements to the content of the article, which can be seen in the revised manuscript.

2.Thank you for your comment. The purpose of Figures 3 and 4 are to clearly present the punch structure to the reader in two-dimension (2D) and three-dimension (3D).

Author Response

Thank you. The English language has been improved.

Reviewer 3 Report

I am recommended this manuscript should be accepted for Publication. Good job to the authors.

Author Response

Thank you.